# POINTSEG: A TRAINING-FREE PARADIGM FOR 3D SCENE SEGMENTATION VIA FOUNDATION MODELS

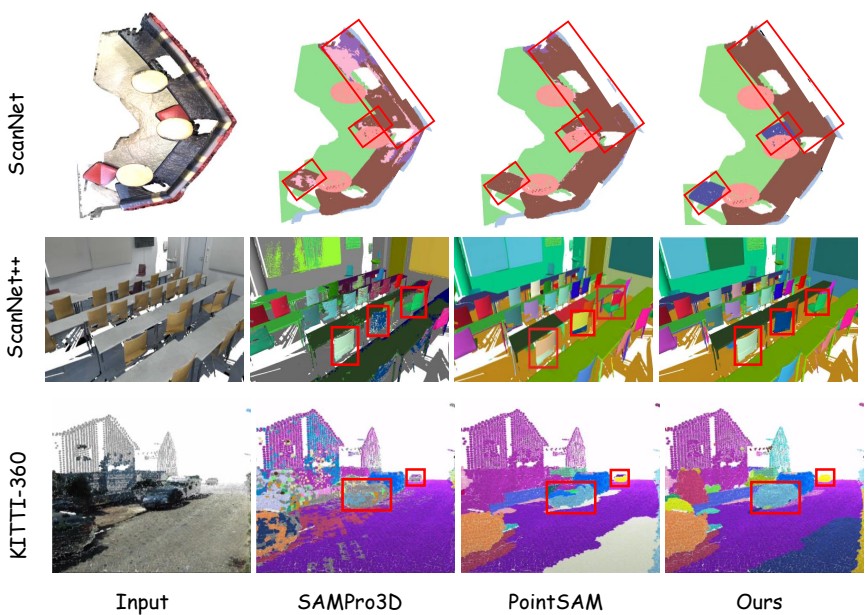

Figure 1: **Qualitative results comparison** on ScanNet, ScanNet++ and KITTI-360 datasets. Compared to the training-based method PointSAM (Zhou et al., 2024) and the training-free method SAMPro3D (Xu et al., 2023), our method can segment objects in 3D scene more completely and accurately.

## ABSTRACT

Recent success of vision foundation models have shown promising performance for the 2D perception tasks. However, it is difficult to train a 3D foundation network directly due to the limited dataset and it remains under explored whether existing foundation models can be lifted to 3D space seamlessly. In this paper, we present PointSeg, a novel training-free paradigm that leverages off-the-shelf vision foundation models to address 3D scene perception tasks. PointSeg can segment anything in 3D scene by acquiring accurate 3D prompts to align their corresponding pixels across frames. Concretely, we design a two-branch prompts learning structure to construct the 3D point-box prompts pairs, combining with the bidirectional matching strategy for accurate point and proposal prompts generation. Then, we perform the iterative post-refinement adaptively when cooperated with different vision foundation models. Moreover, we design a affinity-aware merging algorithm to improve the final ensemble masks. PointSeg demonstrates impressive segmentation performance across various datasets, all without training. Specifically, our approach significantly surpasses the state-of-the-art specialist training-free model by 16.3%, 14.9%, and 15% mAP on ScanNet, ScanNet++, and KITTI-360 datasets, respectively. On top of that, PointSeg can incorporate with various foundation models and even surpasses the specialist training-based methods by 5.6%-8% mAP across various datasets, serving as an effective generalist model.

# 1 INTRODUCTION

3D scene segmentation plays a vital role in many applications, such as autonomous driving, augmented reality and room navigation. To tackle the challenges in 3D scene segmentation, most of the previous methods (Kundu et al., 2020; Jiang et al., 2020; Rozenberszki et al., 2022; Kolodiazhnyi et al., 2024; Liang et al., 2021; Schult et al., 2023; Sun et al., 2023; Vu et al., 2022) are supervised and heavily rely on precise 3D annotations, which means that they lack the zero-shot capability. Despite recent efforts (Takmaz et al., 2023; Huang et al., 2023; Yin et al., 2023; He et al., 2024) have attempted to explore the zero-shot 3D scene understanding, these approaches either require 3D mask pre-trained networks or domain-specific data training. Consequently, the ability of domain transfer to unfamiliar 3D scenes continues to pose significant challenges.

Looking around in the 2D realm, vision foundation models (VFMs) (Radford et al., 2021; Jia et al., 2021; Oquab et al., 2023; He et al., 2022) have exploded in growth, attributed to the availability of large-scale datasets and computational resources. And they have demonstrated exceptional generalization capabilities in zero-shot scenarios, along with multifunctional interactivity when combined with human feedback. Most recently, Segment Anything Model (SAM) (Kirillov et al., 2023) has managed to attain remarkable performance in class-agnostic segmentation by training on the SA-1B dataset. Then it triggers a series of applications in various tasks and improvements in various aspects (Zhang et al., 2023b; Zou et al., 2024; Xiong et al., 2023; Zhao et al., 2023; Zhang et al., 2023a). Inspired by this, a natural idea is to also train a foundation model in 3D space. However, this has been hindered by the limited scale of 3D data and the high cost of 3D data collection and annotation (Goyal et al., 2021; Chang et al., 2015). Considering this, we ask: *Is it possible to explore the use of VFMs to effectively tackle a broad spectrum of 3D perception tasks without training, e.g., 3D scene segmentation ?*

Following this paradigm, some works have made some early attempts. One line focuses on segmenting 2D frame accurately with different scene deconstruction strategies (Yang et al., 2023b; Yin et al., 2024; Guo et al., 2023a). Another line tries to learn high-quality 3D points to prompt the SAM by using the projection from 3D to 2D (Xu et al., 2023). Though effective, none of these methods essentially acknowledge the facts of 3D scene segmentation in three challenging aspects: (i) 3D prompts are naturally prior to the one in the 2D space, which should be carefully designed rather than a simple projection, (ii) the initial segmentation mask from multiple views might include rough edges and isolated background noises, (iii) local adjacent frames maintain the global consensus, which might be overlooked during the merging process.

To address these challenges, we present PointSeg, a novel perception framework that effectively incorporates different foundation models for tackling the 3D scene segmentation task without training. The key idea behind PointSeg is to learn accurate 3D point-box prompts pairs to enforce the off-the-shelf foundation models and fully unleash their potential in 3D scene segmentation tasks with three effective components. First, we construct a two-branch prompts learning structure to acquire the 3D point prompts and 3D box prompts respectively. The 3D point prompts are derived from localization abilities of PointLLM (Xu et al., 2024) to provide more explicit prompts in the form of points and 3D box prompts come from the 3D detectors (Shen et al., 2024; Wu et al., 2023b). Considering that the naive prompts could result in fragmented false-positive masks caused by matching outliers, we propose the bidirectional matching strategy for the generation of accurate point-box prompt pairs. Furthermore, when incorporated with different 2D vision segmentation foundation models, such as SAM 2 (Ravi et al., 2024), our approach involves the iterative post-refinement to eliminate the coarse boundaries and isolated instances of background noise adaptively. Finally, with the primary aim of segmenting all points within the 3D scene, we employ the affinity-aware merging algorithm to capture pairwise similarity scores based on the 2D masks generated by the vision segmentation foundation models.

Comprehensive experiments on ScanNet (Dai et al., 2017), ScanNet++ (Yeshwanth et al., 2023), and KITTI-360 (Liao et al., 2022) demonstrate the superior generalization of the proposed PointSeg, surpassing previous specialist training-free model by 14.9%-16.3% mAP and specialist training-based methods by 5.6%-8% mAP across different datasets, all without training on domain-specific data. Remarkably, our zero-shot approach yields superior results in comparison to fully-supervised PointSAM (Zhou et al., 2024) trained on synthetic datasets, thereby emphasizing the effectiveness of PointSeg in the segmentation of intricate 3D scene. Moreover, we incorporate different foundation

models, *i.e.*, SAM 2 (Ravi et al., 2024), SAM (Kirillov et al., 2023), FastSAM (Zhao et al., 2023), MobileSAM (Zhang et al., 2023a), and EfficientSAM (Xiong et al., 2023), into our pipeline, and the performance gain shows that enhancements on 2D images can be seamlessly translated to improve 3D results. We summarize the contributions of our paper as follows:

- We present PointSeg, a novel framework for exploring the potential of leveraging various vision foundation models in tackling 3D scene segmentation task, without training or finetuning with 3D data.

- We design PointSeg as a two-branch prompts learning structure, equipped with three key components, *i.e.*, bidirectional matching based prompts generation, iterative post-refinement and affinity-aware merging, which can effectively unleash the ability of vision foundation models to improve the 3D segmentation quality.

- PointSeg outperforms previous specialist training-based and training-free methods on 3D segmentation task by a large margin, which demonstrates the impressive performance and powerful generalization when incorporated with various foundation models.

## 2 RELATED WORK

**Closed-set 3D Segmentation.** Considering the point clouds in 3D space, 3D semantic segmentation task aims to predict a specific category towards the given point (Graham et al., 2018; Hu et al., 2020; Kundu et al., 2020; Li et al., 2018; Qi et al., 2017; Rozenberszki et al., 2022; Wang et al., 2019; Xu et al., 2018; Wang et al., 2015; Zhang et al., 2023c). 3D instance segmentation task broadens this concept by pinpointing separate entities within the same semantic class and bestowing unique masks upon each object instance (Choy et al., 2019; Fan et al., 2021; Han et al., 2020; Hou et al., 2019; Engelmann et al., 2020; Hui et al., 2022; Jiang et al., 2020; Kolodiazhnyi et al., 2024; Liang et al., 2021; Schult et al., 2023; Sun et al., 2023; Vu et al., 2022; Lahoud et al., 2019; Yang et al., 2019). Among them, Mask3D (Schult et al., 2023) designs a transformer-based network to build the 3D segmentation network and achieves the state-of-the-art performance. Although Mask3D has made significant progress, like previous supervised learning methods, it still necessitates a substantial volume of 3D annotated data for network training. This limitation impedes the method's generalization to open-world scenarios that include new objects from unseen categories. Furthermore, the collection of annotated 3D data is not only costly but sometimes unfeasible due to privacy concerns. Our framework, however, aspires to directly leverage the intrinsic zero-shot potential of SAM for 3D scene segmentation, thereby negating the necessity for further model training.

**Open-set 3D Segmentation.** Inspired by the success of 2D open-vocabulary segmentation methods (Ghiasi et al., 2022; Liang et al., 2023), a series of works (Ding et al., 2023; Huang et al., 2023; Takmaz et al., 2023; Peng et al., 2023; He et al., 2024; Jiang et al., 2022) have dived to explore the potential of 3D open-vocabulary scene understanding. OpenMask3D (Takmaz et al., 2023) predicts 3D instance masks with the per-mask feature representations, which can be used for querying instances based on open-vocabulary concepts. OpenIns3D (Huang et al., 2023) employs a Mask-Snap-Lookup scheme to learn class-agnostic mask proposals and generate synthetic scene-level images at multiple scales. On the other hand, the interpolation capabilities of NeRFs (Mildenhall et al., 2021) are applied to integrate language with the CLIP feature by LERF (Kerr et al., 2023) and DFF (Kobayashi et al., 2022). And OR-NeRF (Yin et al., 2023) empowers users to segment an object by clicking and subsequently eliminate it from the scene. Although they have achieved encouraging instance segmentation results on indoor scenes with objects similar to the training data, these methods demonstrate a failure in complex scenes with fine-grained objects. In this study, we eliminate the reliance on a pre-trained 3D mask proposal network and instead focus directly on how to leverage the segmentation results of SAM to generate fine-grained 3D masks for 3D scenes.

**Segment Anything Model in 3D.** The emergence of the Segment Anything Model (Kirillov et al., 2023; Ravi et al., 2024) have triggered a revolution in the field of 2D segmentation. Having been trained on the extraordinary SA-1B dataset, SAM has garnered a vast amount of knowledge, equipping it to effectively segment unfamiliar images without additional training. Followed by SAM, several works (Zhang et al., 2023b; Zou et al., 2024; Xiong et al., 2023; Zhao et al., 2023; Zhang et al., 2023a; Liu et al., 2023) have attempted to accelerate or customize the original SAM from different

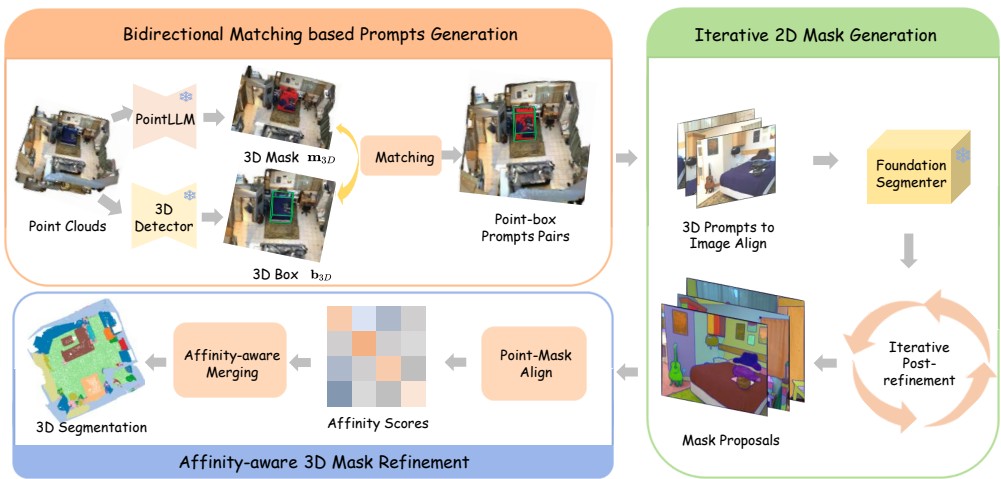

Figure 2: **Overview of our proposed PointSeg**. Our framework requires no training and aims to segment anything in 3D scene via three distinct stages: Bidirectional Matching based Prompts Generation, Iterative 2D Mask Generation, and Affinity-aware 3D Mask Refinement.

aspects. Recognizing the exceptional capabilities of SAM, various recent research initiatives are working diligently to incorporate SAM into 3D scene segmentation task. Several works (Yang et al., 2023b; Yin et al., 2024; Guo et al., 2023a) attempt to segment each frame individually or constructs a graph based on the superpoints to lift the segmentation results to 3D space. However, these methods designate pixel prompts that are specific to each frame but do not synchronize across frames. This causes inconsistencies in segmentation across frames and produces substandard 3D segmentation results. Different from these 2D-to-3D lifting methods, SAMPro3D (Xu et al., 2023) attempts to locate 3D points in scenes as 3D prompts to align their projected pixel prompts across frames. Albeit effective, simply connecting 3D points to 2D space through projection is still too rough for complex scenes. In this paper, we propose a two-branch prompts learning structure towards accurate 3D prompts generation.

## 3 METHOD

As illustrated in Figure 2, we build a training-free 3D scene segmentation framework based on the off-the-shelf foundation models. Our PointSeg consists of three parts: 1) Bidirectional Matching based Prompts Generation (BMP) (Section 3.1). Given the reconstructed scene point cloud $\mathcal{P} = \{\mathbf{p}\}$ together with a set of posed RGB-D images $\{I_m\}_{m=1}^M$, PointSeg first employs a two-branch prompts learning structure to acquire the 3D mask $\mathbf{m}_{3D}$ and box $\mathbf{b}_{3D}$ prompts, respectively. And the final point-box prompts pairs are obtained by the further bidirectional matching. These prompts serve as inputs to 2D vision foundation models, such as SAM 2 (Ravi et al., 2024), after aligning with the pixels in 2D images. 2) Iterative 2D Mask Generation (Section 3.2). Then, we perform Iterative Post-refinement (IPR) to enable the generation of mask proposals adaptively. 3) Affinity-aware 3D Mask Refinement (Section 3.3). We calculate the affinity scores between the points generated in the point-box pairs and the mask proposals, followed by the Affinity-aware Merging (AM) (see Algorithm 2) to obtain the final 3D segmentation masks.

### 3.1 BIDIRECTIONAL MATCHING BASED PROMPTS GENERATION

Towards the generation of accurate 3D prompts, we dive into the exploration of the intrinsic characteristics of 3D data and design a two-branch prompts learning structure. The central concept of our approach entails identifying 3D points and boxes within scenes, serving as inherent 3D prompts, and aligning their projected pixel prompts across various frames. This ensures consistency across frames, both in terms of pixel prompts and the masks predicted by the segmenter.

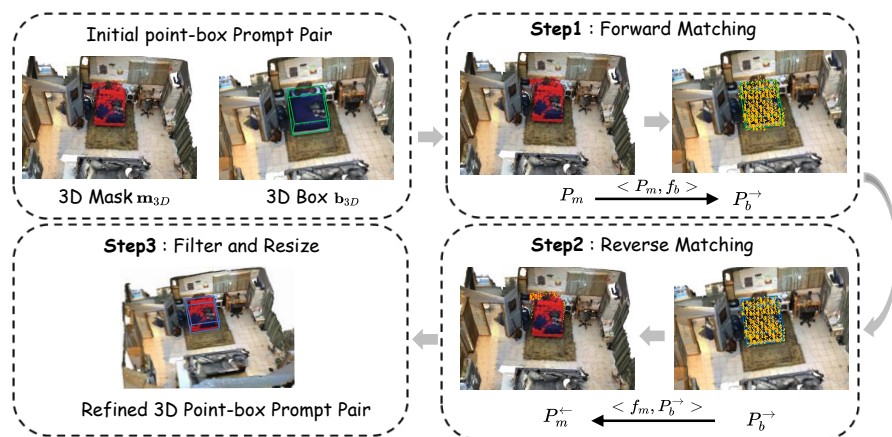

Figure 3: **Illustration of the proposed bidirectional matching**, which consists of three steps: Forward matching, Reserve matching, and Filter and Resize.

**Two-branch Prompts Generation.** Inspired by the strong ability of semantic understanding of some works (Xu et al., 2024; Zhang et al., 2022; Zhu et al., 2023) in 3D scene, we first employ PointLLM (Xu et al., 2024) for localization and rough segmentation in the upper branch in BMP of Figure 2. Given the point cloud $\mathcal{P} \in \mathbb{R}^{N \times 3}$, we also apply realistic projection to generate the different $S$ views, using the zero-initialized 3D grid $G \in \mathbb{R}^{H \times W \times D}$, where $H/W$ denote the spatial resolutions and $D$ represents the depth dimension vertical to the view plane. For each view, the normalized 3D coordinates of the input point cloud $\mathbf{p} = (x, y, z)$ in a voxel in the grid can be denoted as

$$G(\lceil sHx \rceil, \lceil sWy \rceil, \lceil Dz \rceil) = z, \qquad (1)$$

where $s \in (0, 1]$ is the scale factor to adjust the projected shape size. Following PointCLIPv2, we further apply the quantize, densify, smooth, and squeeze operations to obtain the projected depth maps $V = \{v_i\}_{i=1}^{S}$. For the textual input, we utilize the large-scale language models (Brown et al., 2020) to obtain a series of 3D-specific descriptions. After feeding the depth maps and texts into their respective encoders, we can obtain the dense visual features $\{f_i\}_{i=1}^{S}$ where $f_i \in \mathbb{R}^{H \times W \times C}$ and the text feature $f_t \in \mathbb{R}^{K \times C}$. Then, we segment different parts of the shape on multi-view depth maps by dense alignment for each view $i$ and average the back-projected logits of different views into the 3D space, formulated as:

$$f_m = average(Proj^{-1}(f_i \cdot f_t^T)), \qquad (2)$$

where $f_m$ is the segmentation logits in 3D space.

Apart from the point-level prompts in the 3D mask, we intend to inquire into how to generate dense region-level prompts to fully unleash the advantages of 3D prompts in the another branch. To enhance the ability to segment regions accurately in the subsequent segmenters, we exploit the localization abilities of 3D detector to provide more explicit prompts in the form of bounding boxes. The point cloud $\mathcal{P}$ is taken as input by the frozen 3D detectors to generate the 3D bounding box $(x, y, z, w, h, l)$ for each category with corresponding proposal features $f_b$, which can be represented as

$$f_b = Det(\mathcal{P}). \qquad (3)$$

**Bidirectional Matching.** Given the coarse segmentation mask and the bounding box, we can already conduct the alignment to 2D images. However, the naive prompts often result in inaccurate and fragmented outcomes, riddled with numerous outliers. Therefore, we design the bidirectional matching strategy to put constraints on the point and box for high quality promptable segmentation.

Considering the extracted features $f_m$ and $f_b$, we compute the region-wise similarity between the two features to discovery the best matching locations

$$< f_m^i, f_b^j > = \frac{f_m^i \cdot f_b^j}{\|f_m^i\| \cdot \|f_b^j\|}, \qquad (4)$$

**Algorithm 1:** Iterative Post-refinement

**Input:** the projected point coordinates $\mathbf{x}$ and box $\mathbf{b}$ in frame $i$, the predefined threshold $\vartheta$
**Output:** the refined mask $M_i$

1: $\Delta \leftarrow \infty$
2: $i \leftarrow 1$
3: $M_0 = Dec_M(\mathbf{x}, \mathbf{b})$
4: **while** $\Delta > \vartheta$ **do**
5: $\quad M_i = Dec_M(\mathbf{x}, \mathbf{b}, M_{i-1})$
6: $\quad \Delta \leftarrow \frac{\sum_{j=1}^{N}(M_{i,j} - M_{i-1,j})}{M_{i-1,j}}$
7: $\quad i \leftarrow i + 1$
8: **end while**
9: **return** $M_i$

**Algorithm 2:** Affinity-aware Merging

**Input:** affinity matrix $A \in \mathbb{R}^{N \times N}$ where $A_{i,j}$ indicates the affinity score between two points $p_i$ and $p_j$, and $N$ is the number of points.
**Output:** mask label $l \in \mathbb{R}^N$.
1: $l \leftarrow 0, id \leftarrow 1$
2: **for** $i \leftarrow 1$ to $N$ **do**
3: $\quad$ **if** $l_i = 0$ **then**
4: $\quad\quad l_i \leftarrow id$
5: $\quad\quad$ **for** each $j$ in neighbors of $i$ **do**
6: $\quad\quad\quad$ **if** $l_j \neq 0$ **then**
7: $\quad\quad\quad\quad$ continue
8: $\quad\quad\quad$ **end if**
9: $\quad\quad\quad R \leftarrow \{p_k | l_k = id\}$
10: $\quad\quad\quad A_{R,p_j} \leftarrow$ region-point score$(R, p_j, A)$
11: $\quad\quad\quad$ **if** $A_{R,p_j} > \tau$ **then**
12: $\quad\quad\quad\quad l_j \leftarrow id, i \leftarrow j, id \leftarrow id + 1$
13: $\quad\quad\quad\quad$ continue
14: $\quad\quad\quad$ **end if**
15: $\quad\quad$ **end for**
16: $\quad\quad id \leftarrow id + 1$
17: $\quad$ **end if**
18: **end for**

where $< f_m^i, f_b^j >$ denotes the cosine similarity between $i$-th mask feature and $j$-th box feature. Ideally, the matched regions should have the highest similarity. Then as illustrated in Figure 3, we propose to eliminate the matching outliers based on the similarity scores in three steps:

- First, we compute the forward similarity $< P_m, f_b >$ between the points in the mask $P_m$ and the box $f_b$. Using this score, the bipartite matching is performed to acquire the forward matched points $P_b^{\rightarrow}$ within the box.

- Then, we perform reverse matching between the matched points $P_b^{\rightarrow}$ and $f_m$ to obtain the reverse matched points $P_m^{\leftarrow}$ within the mask using the reverse similarity $< f_m, P_b^{\rightarrow} >$.

- Finally, we resize the box according to the points in the forward sets if the corresponding reverse points are not within mask, denoted as $\hat{P}_b = \{\mathbf{p}_b^i \in P_b^{\rightarrow} | \mathbf{p}_m^i \ in \ \mathbf{m}_{3D}\}$. Similarly, we adjust points in the mask by filtering out the points in the reverse set if the corresponding forward points are not within the box, denoted as $\hat{P}_m = \{\mathbf{p}_m^i \in P_m^{\leftarrow} | \mathbf{p}_b^i \ in \ \mathbf{b}_{3D}\}$.

In this way, we can form the point-box pairs with a new set of points in the mask and a different box with new size, which are feed into the further segmentation module.

## 3.2 ITERATIVE 2D MASK GENERATION

Conditioned on the reorganized point-box pairs, we then make the alignment between the 3D prompts pairs and the 2D images. In particular, given a point $\mathbf{p}$ in the prompts pairs with the camera intrinsic matrix $I_i$ and world-to-camera extrinsic matrix $E_i$, the corresponding pixel projection $\mathbf{x}$ can be calculated by

$$\mathbf{x} = (u, v) = I_i \cdot E_i \cdot \tilde{\mathbf{p}}, \tag{5}$$

where $\tilde{\mathbf{p}}$ is the homogeneous coordinates of $\mathbf{p}$. Similarly, the corresponding projected box across images can be denoted as $\mathbf{b} = (u, v, h, w)$.

The vanilla segmentation model, such as SAM 2 (Ravi et al., 2024), accepts various inputs such as pixel coordinates, bounding boxes or masks and predict the segmentation area associated with each prompt. Hence, we feed the projected 2D point-box pairs calculated before into the foundation segmenters.

**Iterative Post-refinement.** Through the above operation, we can obtain 2D segmentation mask on all frames from the decoder, which however, might include rough edges and isolated background noises. For further refinement, we iteratively feed the mask back into the decoder $Dec_M$ for the adaptive post-processing. As illustrated in Algorithm 1, we first obtain the 2D mask $M_0$ by feeding the 2D point-box pairs into the SAM-based decoder. Then we prompt the decoder additionally with this mask along with these projected prompt pairs to obtain the next mask. And the initial value $\Delta$ to record change is set to infinity. In each subsequent iteration, we calculate the change ratio between

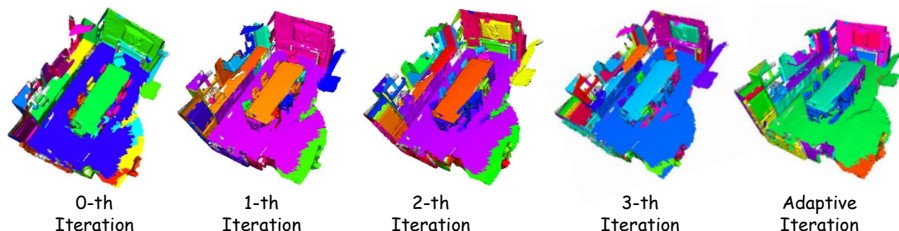

| 0-th Iteration | 1-th Iteration | 2-th Iteration | 3-th Iteration | Adaptive Iteration |

Figure 4: **Different Iteration strategies.** Without adaptive iteration, the segmentation results can be sensitive to the number of fixed iteration steps. But the performance of adaptive iteration is proved to be more effective.

the two adjacent masks and compare it with our predefined threshold $\vartheta$, which is set to 5% by default. We repeat this iterative process until the change value falls below the threshold adaptively. It is worth noting that we have also tried the method of fixed iteration steps, but this adaptive iteration method have proved to be more effective which is shown in Figure 4 and Table 6.

### 3.3 AFFINITY-AWARE 3D MASK REFINEMENT

After previous procedures, we have obtained the final set of 2D segmentation masks across frames. With the ultimate goal of segmenting all points in the 3D scene, we employ the affinity-aware merging algorithm to generate the final 3D masks.

**Affinity-aware Merging.** Based on the $i$-th projected point-box pair on the $m$-th image and their 2D image segmentation mask, we compute the normalized distribution of the mask labels, denoted as $\mathbf{d}_{i,m}$. The affinity score between two projected points in the $m$-th image can be computed by the cosine similarity between the two distributions, which can be represented as:

$$A_{i,j}^m = \frac{\mathbf{d}_{i,m} \cdot \mathbf{d}_{j,m}}{\|\mathbf{d}_{i,m}\| \cdot \|\mathbf{d}_{j,m}\|}. \tag{6}$$

The final affinity score across different images can be computed by the weighted-sum:

$$A_{i,j} = \frac{\sum_{m=1}^{M} \alpha_{i,j}^m \cdot A_{i,j}^m}{\sum_{m=1}^{M} \alpha_{i,j}^m}, \tag{7}$$

where $\alpha_{i,j}^m \in (0,1)$ denotes whether it is visible in the images.

Further, we utilize the designed affinity-aware merging algorithm to gradually merge 3D masks using the computed affinity matrix. As illustrated in Algorithm 2, the algorithm works on an affinity matrix representing the affinity scores between pairs of points. The goal is to assign labels to these points based on their affinities.

We start by initializing labels and an identifier. It then iterates over each point. If a point hasn't been labeled yet, it gets the current identifier. The algorithm then checks the point's neighbors. If a neighbor is unlabeled, it calculates an affinity score between the set of already labeled points and the current neighbor. If this score surpasses a certain threshold, the neighbor is labeled with the current identifier, the identifier is incremented, and the algorithm continues with this neighbor as the current point. The region-point merging is similar to Equation 7, which is computed as the weighted average between the current point and the points inside the region. The whole process repeats until all points have been labeled, effectively grouping points based on their mutual affinities.

## 4 EXPERIMENT

### 4.1 SETUP

**Baselines.** We compare our approach with both training-based and training-free baselines. For training-based comparison, we select the state-of-the-art transformer-based method PointSAM (Zhou

Table 1: **Results of 3D segmentation on ScanNet, ScanNet++, and KITTI-360 datasets**. We report the mAP and AP scores on the three datasets. Best results are highlighted in **bold**.

| Method | Type | ScanNet | | | ScanNet++ | | | KITTI-360 | | |
|---|---|---|---|---|---|---|---|---|---|---|
| | | mAP | $AP_{50}$ | $AP_{25}$ | mAP | $AP_{50}$ | $AP_{25}$ | mAP | $AP_{50}$ | $AP_{25}$ |
| *With Training* | | | | | | | | | | |
| SAM-graph (Guo et al., 2023a) | *specialist model* | 15.1 | 33.3 | 59.1 | 12.9 | 25.3 | 43.6 | 14.7 | 28.0 | 43.2 |
| Mask3D (Schult et al., 2023) | *specialist model* | 26.9 | 44.4 | 57.5 | 8.8 | 15.0 | 22.3 | 0.1 | 0.4 | 4.2 |
| OpenDAS (Yilmaz et al., 2024) | *specialist model* | 28.3 | 49.6 | 66.2 | 19.2 | 35.5 | 52.6 | 20.1 | 32.4 | 52.2 |
| PointSAM (Zhou et al., 2024) | *specialist model* | 32.9 | 56.4 | 72.5 | 25.8 | 38.0 | 59.3 | 25.1 | 38.4 | 56.2 |
| *Training-free* | | | | | | | | | | |
| SAM3D (Yang et al., 2023b) | *specialist model* | 13.7 | 29.7 | 54.5 | 8.3 | 17.5 | 33.7 | 6.3 | 16.0 | 35.6 |
| SAI3D (Yin et al., 2024) | *specialist model* | 18.8 | 42.5 | 62.3 | 17.1 | 31.1 | 49.5 | 16.5 | 30.2 | 48.6 |
| SAMPro3D (Xu et al., 2023) | *specialist model* | 22.2 | 45.6 | 65.7 | 18.9 | 33.7 | 51.6 | 17.3 | 31.1 | 49.6 |
| **PointSeg (Ours)** | *generalist model* | **38.5** | **63.6** | **82.1** | **33.8** | **49.1** | **67.2** | **32.3** | **47.2** | **66.5** |

Table 2: **Results of integrating with different segmentation models** on ScanNet, ScanNet++, and KITTI-360 datasets.

| Method | ScanNet | | | ScanNet++ | | | KITTI-360 | | |
|---|---|---|---|---|---|---|---|---|---|
| | mAP | $AP_{50}$ | $AP_{25}$ | mAP | $AP_{50}$ | $AP_{25}$ | mAP | $AP_{50}$ | $AP_{25}$ |
| + SAM 2 (Ravi et al., 2024) | **38.5** | **63.6** | **82.1** | **33.8** | **49.1** | **67.2** | **32.3** | **47.2** | **66.5** |
| + SAM (Kirillov et al., 2023) | 36.3 | 60.2 | 79.3 | 31.2 | 46.5 | 64.8 | 29.9 | 44.5 | 63.3 |
| + MobileSAM (Zhang et al., 2023a) | 26.2 | 49.8 | 68.3 | 19.6 | 36.4 | 55.2 | 20.6 | 34.6 | 53.3 |
| + FastSAM (Zhao et al., 2023) | 26.9 | 50.8 | 69.1 | 20.5 | 37.7 | 56.5 | 21.2 | 35.8 | 54.5 |
| + EfficientSAM (Xiong et al., 2023) | 33.5 | 57.2 | 75.8 | 27.8 | 43.6 | 62.4 | 27.5 | 41.7 | 60.6 |

et al., 2024). For training-free methods, we compare against the 2D-to-3D lifting methods (Yang et al., 2023b; Xu et al., 2023) and the 3D-to-2D projection methods (Xu et al., 2023) respectively. For Implementation, we apply the V-DETR (Shen et al., 2024) and VirConv (Wu et al., 2023b) as the indoor and outdoor 3D detector.

## 4.2 MAIN RESULTS

**Comparisons with the state-of-the-art Methods.** We compare the segmentation results on Scan-Net, ScanNet++, and KITTI-360 datasets, covering both indoor and outdoor scenes. As shown in Table 1, comparing to previous training-free methods, our PointSeg obtains 16.3% mAP, 18% $AP_{50}$, and 16.4% $AP_{25}$ performance gains on ScanNet. On the more challenging indoor dataset ScanNet++, our method still obtains 14.9% mAP, 15.4% $AP_{50}$, and 15.6% $AP_{25}$ improvements. Furthermore, when evaluating the performance of our PointSeg on outdoor KITTI-360, our method still surpasses corresponding zero-shot method by 15% mAP, 16.1% $AP_{50}$, and 16.9% $AP_{25}$, respectively. In this regard, our PointSeg demonstrates superior generalization ability to complex 3D scenarios.

Notably, when compared to previous training-based methods, PointSeg outperforms PointSAM (Zhou et al., 2024) by 5.6%-8%, 7.2%-11.1%, 7.9%-10.3% in terms of mAP, $AP_{50}$, and $AP_{25}$ across various datasets. This further demonstrates the robustness and effectiveness of our approach in 3D segmentation task.

Table 3: **Ablations results of different 3D point models**.

| Method | mAP | $AP_{50}$ | $AP_{25}$ |
|---|---|---|---|
| + PointCLIP (Zhang et al., 2022) | 32.3 | 56.9 | 76.2 |
| + PointCLIPv2 (Zhu et al., 2023) | 34.6 | 58.2 | 77.6 |
| + ULIP (Xue et al., 2023) | 34.1 | 57.8 | 77.3 |
| + ULIPv2 (Xue et al., 2024) | 35.7 | 59.1 | 78.1 |
| + PointBIND (Guo et al., 2023b) | 36.1 | 59.6 | 78.8 |
| + PointLLM (Xu et al., 2024) | 36.3 | 60.2 | 79.3 |

Table 4: **Ablations results of different 3D detectors**.

| Method | mAP | $AP_{50}$ | $AP_{25}$ |
|---|---|---|---|
| *Indoor* | | | |
| + V-DETR (Shen et al., 2024) | 36.3 | 60.2 | 79.3 |
| + Swin3d (Yang et al., 2023a) | 35.6 | 59.7 | 78.1 |
| + CAGroup3D (Wang et al., 2022) | 34.2 | 58.1 | 77.2 |
| *Outdoor* | | | |
| + Virconv (Wu et al., 2023b) | 36.3 | 60.2 | 79.3 |
| + TED (Wu et al., 2023a) | 33.5 | 56.8 | 75.1 |
| + LoGoNet (Li et al., 2023) | 33.1 | 56.2 | 74.8 |

Table 5: **Ablations of main components in our framework.**

| BMP | IPR | AM | mAP | AP$_{50}$ | AP$_{25}$ |
|---|---|---|---|---|---|
| - | - | - | 16.2 | 40.6 | 60.3 |
| - | - | ✓ | 28.6 | 50.9 | 69.7 |
| - | ✓ | - | 32.5 | 55.2 | 74.3 |
| ✓ | - | - | 29.5 | 51.9 | 71.2 |
| ✓ | ✓ | - | 33.9 | 57.8 | 76.5 |
| ✓ | - | ✓ | 31.7 | 53.3 | 75.1 |
| - | ✓ | ✓ | 34.6 | 58.2 | 77.5 |
| ✓ | ✓ | ✓ | 36.3 | 60.2 | 79.3 |

Table 6: **Ablation of iterative post-refinement.**

| Strategy | | mAP | AP$_{50}$ | AP$_{25}$ |
|---|---|---|---|---|
| Iter | 0 | 30.3 | 54.8 | 73.8 |
| | 1 | 31.9 | 55.6 | 74.1 |
| | 2 | 32.6 | 56.8 | 75.7 |
| | 3 | 34.7 | 58.5 | 77.8 |
| | 4 | 32.3 | 56.9 | 75.5 |
| | 5 | 30.4 | 54.1 | 73.2 |
| Adaptive | | 36.3 | 60.2 | 79.3 |

**Qualitative Results.** The representative quantitative segmentation results of our proposed PointSeg on the three datasets are shown in Figure 1 and Figure A1. We also present the quantitative results of the state-of-the-art training-based methods PointSAM (Zhou et al., 2024) and training-free method SAMPro3D (Xu et al., 2023). We can observe that PointSAM and SAMPro3D often mistakenly segment an object into two different objects and exhibit poor performance in segmenting relative objects lacked spatial structure. Our PointSeg can handle complex scenes and is capable of generating clean segments on objects of small size, further underscoring the effectiveness of our approach.

### 4.3 ABLATION STUDY

In this section, we conduct extensive ablation studies on ScanNet to show the effectiveness of each component and design. Unless otherwise specified, SAM (Kirillov et al., 2023) is used as 2D segmentation foundation model for ablation studies by default.

**Different Foundation Models.** *i)* Apart the basic segmentation foundation model SAM 2 (Ravi et al., 2024), we also integrate different segmentation foundation model, *i.e.*, SAM (Kirillov et al., 2023), MobileSAM (Zhang et al., 2023a), FastSAM (Zhao et al., 2023), and EfficientSAM (Xiong et al., 2023), into our framework. As shown in Table 2, PointSeg demonstrates consistent performance improvements among different datasets, where the original SAM 2-based result performs best. This is consistent with the relative results of these methods in 2D segmentation. *ii)* In Table 3, we change the model for localization of point prompts. The results demonstrate that more accurate points for 3D prompts can indeed contribute to the performance gain. *iii)* In Table 4, the improvement of different 3D detectors will also bring about the improvement of the performance of our method. These results suggest that our framework can serve as a foundational structure, capable of integrating a variety of fundamental models. And the enhancements observed in these models can be smoothly translated to 3D space, thereby augmenting the overall performance.

**Main Components.** Further, we explore the effects of bidirectional matching based prompts generation (BMP), iterative post-refinement (IPR) and affinity-aware merging (AM). We illustrate the importance of different components by removing some parts and keeping all the others unchanged. The baseline setting is to use points and boxes independently as 3D prompts. And the masks from the 2D segmentation model decoder are used to merge the final 3D masks, without any post-refinement. The merging strategy is same as the adjacent frame merging in (Yang et al., 2023b). The results of components ablations are shown in Table 5. We observe that using the iterative post-refinement strategy leads to a noticeable increase in performance, which demonstrates the necessity of refining the initial 2D masks. The performance degradation caused by the absence of bidirectional matching proves that the constraints between the point and box prompts can indeed help to generate the accurate point-box pairs. And the performance drop without the affinity-aware merging shows that the affinity score is indeed useful to link the point and the masks.

**Iterative Post-refinement.** As mentioned before, when performing the post-refinement during the initial 2D mask generation, we have tried different strategies of fixed numbers method and adaptive iteration method. As shown in Table 6, the six rows in the middle represent the method to use a fixed number of iterations and the last row is the adaptive iteration. We can observe that as the number of iterations increases, the corresponding AP value also becomes higher compared to no iteration, which

Table 7: **Ablation of bidirectional matching**. *no* means no matching.

| Strategy | mAP | AP$_{50}$ | AP$_{25}$ |
|---|---|---|---|
| *no* | 22.2 | 49.6 | 67.3 |
| forward | 30.9 | 55.5 | 73.5 |
| reverse | 32.6 | 56.7 | 75.5 |
| bidirectional | 36.3 | 60.2 | 79.3 |

Table 8: **Ablation of affinity-aware merging algorithm**.

| Merging | mAP | AP$_{50}$ | AP$_{25}$ |
|---|---|---|---|
| BM | 27.6 | 53.5 | 73.7 |
| PM | 31.5 | 55.7 | 76.2 |
| IDM | 32.1 | 56.2 | 76.8 |
| AM | 36.3 | 60.2 | 79.3 |

Table 9: **Ablation of 3D point pre-trained models from the two branches**.

| Strategy | mAP | AP$_{50}$ | AP$_{25}$ |
|---|---|---|---|
| PointClip only | 30.2 | 51.2 | 72.7 |
| 3D detector only | 30.1 | 51.3 | 72.6 |
| combine(w/o matching) | 34.6 | 58.2 | 77.5 |
| combine(w/ matchfing) | 36.3 | 60.2 | 79.3 |

shows that the obtained mask is also more accurate. In the method with a fixed number of iterations, the results reach the highest in the third iteration, but are still lower than those in the adaptive iteration method. This largely illustrates the effectiveness of the iterative post-processing method in generating more accurate masks, and also reveals that the adaptive iteration is more beneficial.

**Different Matching Strategy.** To validate the effect of different matching method, we explore the effects of the forward matching and the reverse matching of the proposed bidirectional matching, as shown in Table 7. Without the guidance from the respective point and box, the naive prompts contain many invalid points and regions, which provide negative prompts for the following segmentation models. Our bidirectional matching improves the performance of forward and reverse matching by 5.4% and 3.7 %, which show the effectiveness of the proposed bidirectional matching strategy.

**Affinity-aware Merging.** In the final mask merging stage, we have also tried other merging algorithm. As shown in Table 8, we compare our affinity-aware merging (AM) with (a) bidirectional merging (BM) from (Yang et al., 2023b), (b) pure merging (PM) without affinity scores, which is simplified from our approach and (c) prompt ID based merging (IDM) from (Xu et al., 2023). With other inferior merging method, the performance drops dramatically which shows the superiority of our proposed affinity-aware merging algorithm in solving the mask merging problems in 3D scene.

Table 10: **Ablation of mask change ratio**.

| Ratio | mAP | AP$_{50}$ | AP$_{25}$ |
|---|---|---|---|
| 1% | 33.6 | 57.5 | 76.6 |
| 3% | 35.2 | 59.3 | 78.7 |
| 5% | 36.3 | 60.2 | 79.3 |
| 8% | 35.8 | 59.7 | 78.1 |
| 10% | 34.6 | 58.3 | 77.9 |
| 15% | 32.2 | 56.6 | 75.5 |

**3D Point Models.** As shown in Table 9, the missing of PointLLM/3D detector causes the performance drop and only the matching of combination performs best.

Table 11: **Inference Speed.**

| module | bidirectional matching | iterative refinement | affinity-aware merging |
|---|---|---|---|
| FPS | 1.53 | 1.05 | 1.96 |

**Mask Change Ratio.** In the iterative post-refinement module, we set a mask change ratio threshold $\vartheta$ as a condition for stopping the iteration. Here, We show the effect of different ratios on the results. As shown in Table 10, the overall results perform best when mask change ratio is set as 5%.

**Inference Speed.** As shown in Table 11, we test the running efficiency on NVIDIA V100 GPU and the detailed FPS of each module. The FLOPs of our method is 4.2G.

## 5 CONCLUSION

In this paper, we present PointSeg, a novel training-free framework integrating off-the-shelf vision foundation models for solving 3D scene segmentation tasks. The key idea is to learn accurate 3D point-box prompts pairs to enforce the off-the-shelf foundation models. Combining the three universal components, *i.e.*, bidirectional matching based prompts generation, iterative post-refinement and affinity-aware merging, PointSeg can effectively unleash the ability of various foundation models. Extensive experiments on both indoor and outdoor datasets demonstrate that PointSeg outperforms prior unsupervised methods and even surpass fully-supervised by a large margin, which reveals the superiority of our model in 3D scene understanding task.

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

APPENDIX

## A DATASETS AND METRICS.

To validate the effectiveness of our proposed PointSeg, we conduct extensive experiments on three popular public benchmarks: ScanNet (Dai et al., 2017), ScanNet++ (Yeshwanth et al., 2023), and KITTI-360 (Liao et al., 2022). ScanNet provides RGBD images and 3D meshes of 1613 indoor scenes. ScanNet++ is a recently released indoor dataset with more detailed segmentation masks, serving as a more challenging benchmark for 3D scenarios. It contains 280 indoor scenes with high-fidelity geometry and high-resolution RGB images. KITTI-360 is a substantial outdoor dataset that includes 300 suburban scenes, which comprises 320k images and 100k laser scans. We evaluate ours segmentation performance with the widely-used Average Precision (AP) score. Following (Schult et al., 2023; Dai et al., 2017), we report AP with thresholds of 50% and 25% (denoted as $AP_{50}$ and $AP_{25}$) as well as AP averaged with IoU thresholds from 50% to 95% with a step size of 5% (mAP).

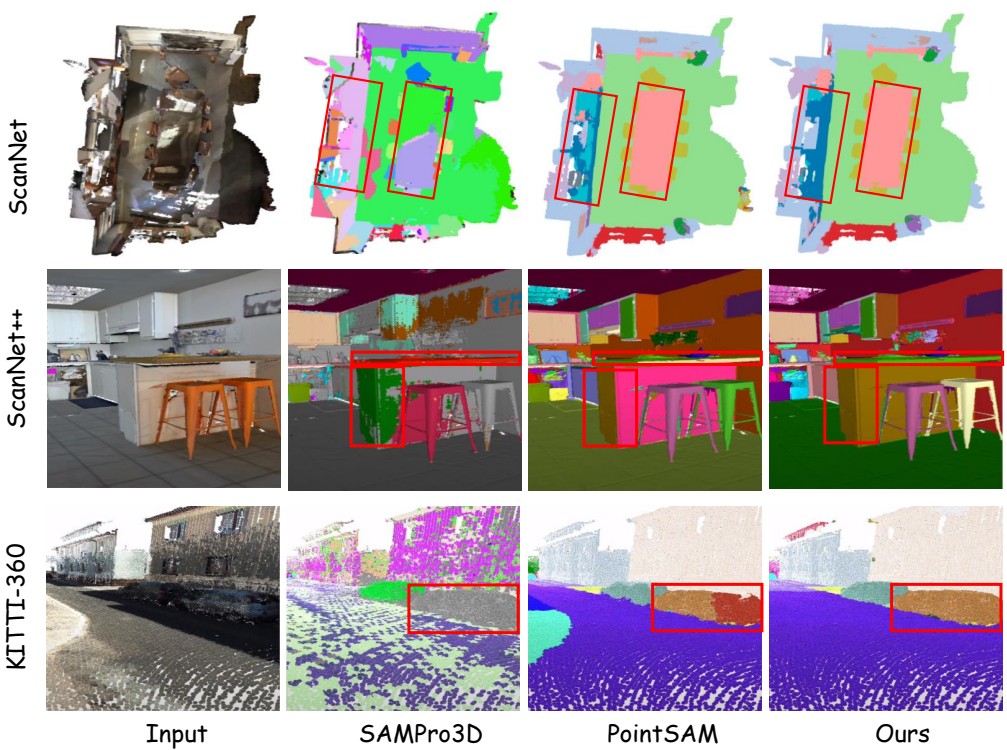

Figure A1: **More qualitative results comparison** on ScanNet, ScanNet++ and KITTI-360 datasets with comparison to the training-based method PointSAM (Zhou et al., 2024) and the training-free method SAMPro3D (Xu et al., 2023).

