# OpenReview forum: "PointSeg: A Training-Free Paradigm for 3D Scene Segmentation via Foundation Models"
_ICLR.cc/2025/Conference — ICLR 2025 Conference Withdrawn Submission_

### Official Review · Reviewer_XWQy · 2024-11-01

**Soundness:** 2
**Presentation:** 1
**Contribution:** 2
**Rating:** 5
**Confidence:** 3

**Summary:**

The paper presents PointSeg, a training-free approach that leverages foundational vision models for 3D scene segmentation. PointSeg uses pretrained models to generate point and bounding box prompts, which are then used to prompt the SAM model for 2D segmentation masks. Through Iterative Post-Refinement, SAM is repeatedly prompted to improve 2D mask accuracy, while Affinity-aware Merging integrates the refined 2D masks into 3D. PointSeg is evaluated on both indoor and outdoor datasets, with extensive ablation studies conducted to validate its design choices.

**Strengths:**

1. The method introduces a novel approach by combining detection bounding boxes with point prompts to guide SAM, ensuring 3D consistency in prompts.
2. The reported quantitative results outperform current SOTA methods.

**Weaknesses:**

1. The description of the Two-Branch Prompt Generation process lacks clarity. Could you clarify how PointLLM is used for rough localization? What specific input prompts are used for PointLLM? Which large-scale language models are used to generate the text input, and with what prompts? Can you provide details on the models used to obtain dense visual and text features?
2. The notation $K$ in line 248 needs clarification. Could you please define the notation $K$ used in line 248? If it refers to the number of classes, how is this determined? Is $K$ an input parameter for the algorithm or derived in some way?
3. The SOTA comparisons are unclear. Can you provide a detailed description of how the comparisons with SOTA models were conducted? Specifically, what prompts were used for promptable segmentation models like PointSAM? If PointSeg's generated prompts were used for other models, please explain this process.
4. The experiment setup is unclear. Could you provide a step-by-step explanation of how the produced masks were aligned with ground truth masks for AP calculation? Given that PointSeg does not generate semantic labels, what method was used to match the segmented instances with ground truth instances?

**Questions:**

Please see weaknesses.

---

### Official Review · Reviewer_q2cX · 2024-11-01

**Soundness:** 3
**Presentation:** 2
**Contribution:** 2
**Rating:** 5
**Confidence:** 3

**Summary:**

The paper presents PointSeg, a training-free method that uses existing 2D vision models for 3D scene segmentation. PointSeg aligns 3D prompts with pixels across frames and employs a two-branch structure with adaptive refinement and merging algorithms. It outperforms state-of-the-art training-free models on several 3D datasets and even surpasses some training-based methods, demonstrating its effectiveness as a generalist model for 3D segmentation.

**Strengths:**

1. PointSeg does not require additional training, making it efficient and easy to deploy across different tasks and datasets.

2. It utilizes off-the-shelf 2D vision foundation models, alleviating the challenge of limited 3D datasets and reducing the need to build a 3D model from scratch.

3. The proposed components are useful based on the ablation.

**Weaknesses:**

1. It is a more engineering-oriented work that lacks novelty, mainly combining various large models rather than being a research-focused work.

2. The paper lacks some relevant references. It uses 2D models to segment 3D scenes, as do the following works. However, this paper does not provide a comparative discussion with them:
[1] GOV-NeSF: Generalizable Open-Vocabulary Neural Semantic Fields. Yunsong Wang, etc. CVPR, 2024.
[2] Lift3D: Zero-Shot Lifting of Any 2D Vision Model to 3D. Mukund Varma T, etc. CVPR, 2024
[3] GNeSF: Generalizable Neural Semantic Fields. Hanlin Chen, etc, NeurIPS, 2023.

3. The paper is unclear in certain aspects. For example, is the scale factor a hyperparameter or a learnable parameter? If it is a hyperparameter, what is the set value?

4. The paper lacks implementation details.

**Questions:**

The questions can be seen under "weakness." Additionally, there is another issue as follows:

1. How long does it take on average to perform inference on a single scene, including the time spent on various large models?

---

### Official Review · Reviewer_Mqx3 · 2024-11-03

**Soundness:** 3
**Presentation:** 2
**Contribution:** 3
**Rating:** 6
**Confidence:** 3

**Summary:**

This paper introduces PointSeg, a novel framework for 3D scene segmentation that operates without the need for training, leveraging pre-trained 2D foundation models. PointSeg employs a unique two-branch prompt generation architecture and a bidirectional matching strategy to project 3D point cloud data into meaningful 2D cues. These cues, aligned with pre-trained 2D models, provide spatially accurate segmentation in 3D spaces. The authors evaluate PointSeg on various datasets, demonstrating its efficacy in achieving high segmentation accuracy across different indoor and outdoor scenes. The primary contributions of the paper are as follows:
i) A Training-Free 3D Segmentation Framework: PointSeg introduces a framework that bypasses the need for 3D model training by leveraging pre-trained 2D models, making it computationally efficient and adaptable to new environments.
ii) Bidirectional Matching-Based Prompt Generation: The proposed two-branch architecture generates both point and box prompts, which are refined through bidirectional matching, ensuring high segmentation accuracy.
iii) Comprehensive Evaluation: The paper presents extensive experimental results, comparing PointSeg’s performance with baseline models and demonstrating its competitive accuracy and adaptability across multiple 3D segmentation tasks.
Overall, PointSeg represents a significant step toward making 3D scene segmentation more accessible and efficient by eliminating the need for training and instead capitalizing on foundation models.

**Strengths:**

PointSeg introduces a novel and efficient approach to 3D segmentation that expands the possibilities for leveraging 2D foundation models in new domains, and its contributions are both methodologically sound and practically valuable. Explicitly,
1) PointSeg pioneers a framework that performs 3D scene segmentation without requiring training or fine-tuning on 3D data, showcasing the untapped potential of visual foundation models in 3D tasks.
2) The dual-branch structure, incorporating bidirectional matching, iterative refinement, and affinity-aware merging, maximizes the capabilities of foundation models, delivering high-quality 3D segmentation results through a versatile prompt-based mechanism.
3) PointSeg achieves outstanding performance on 3D segmentation tasks, surpassing both training-based and training-free methods, and demonstrates strong adaptability across different foundation models, highlighting its broad applicability and effectiveness.

**Weaknesses:**

1) Novelty: The framework heavily relies on the quality and appropriateness of the selected pre-trained 2D models. If these models do not generalize well to the specific characteristics of 3D scenes, it may hinder PointSeg's performance. The paper could benefit from discussing potential limitations arising from this dependency.
2)  Interaction: Although the three key components (bidirectional matching, iterative refinement, and affinity-aware merging) are introduced, the paper could provide more in-depth explanations and analyses of how each component contributes to overall performance. A more detailed exploration of their interactions and potential trade-offs would enhance the understanding of their roles within the framework.
3) Generalization: The paper only tests on the ScanNet, ScanNet++, and KITTI-360 datasets, lacking validation on a more diverse range of scenes, such as more complex 3D environments or dynamic scenarios (e.g., Waymo, nuScenes). This limitation may affect the universality of the conclusions drawn.

**Questions:**

1) Experimental setups：The paper does not provide a detailed explanation of how hyperparameters for each module in the model (such as iteration count, thresholds, etc.) are set. The absence of this information may affect the reproducibility of the results.
2) Theoretical validation: While the paper presents some mathematical derivations, further elaboration and detail would enhance the reader's understanding of the model's operational mechanisms. For instance, the absence of a detailed analysis of the interrelationships among key components may affect the depth and comprehensiveness of the theoretical explanations. Providing additional derivations or clarifications on how these components interact and contribute to the overall performance of the model would strengthen the theoretical foundation of the paper and improve its clarity for the audience.
3) Generalization performance: The paper's testing only covers the ScanNet, ScanNet++, and KITTI-360 datasets, lacking validation on a more diverse range of scenes. Furthermore, what is the model's generalization performance on point cloud data collected from real-world sources, such as data obtained from mobile devices or drones?

---

### Official Review · Reviewer_2baF · 2024-11-04

**Soundness:** 3
**Presentation:** 2
**Contribution:** 2
**Rating:** 5
**Confidence:** 3

**Summary:**

This paper introduces PointSeg, a training-free framework designed to leverage existing 2D vision foundation models for 3D scene segmentation tasks. PointSeg achieves this by generating precise 3D prompts, allowing it to match corresponding pixels across frames and segment objects in 3D space effectively. The approach is composed of a two-branch prompt-learning structure, bidirectional matching for point and proposal prompts, adaptive post-refinement with foundation models, and an affinity-aware merging algorithm for enhanced mask accuracy. PointSeg demonstrates substantial improvements over existing training-free models, surpassing state-of-the-art performance on datasets like ScanNet and KITTI-360  and even outperforming some training-based models.

**Strengths:**

Extensive experiments are conducted to demonstrate the effectiveness of the proposed method. The results are promising.  The paper is well-organized. It clearly reflects the significant effort put into building a complete pipeline and carefully designing and conducting the experiments.

**Weaknesses:**

1. Though the organization of the paper is okay, I think the method is not easy to follow. First, the writing is not clear enough. For example, the definition in Equation (1) is weird. How to understand the ‘=z’?
2. How to understand Equation (4)? What is Det() in Equation (3)? Why can the proposal feature $f_b$ and the segmentation logits $f_m$ be used to calculate cosine similarity? This part is confusing.
3. The Affinity-aware Merging algorithm seems to be a depth-first algorithm similar to the union-find set. I think Alg.2 is confusing. In Line 12 of Alg.2, if $A_{R, p_j} > \tau$, then the same id is assigned to this point j. However, after that, I guess it should consider the neighbors of j and check whether they belong to the same group of j. And that is why we have i <- j . However, the following id <- id + 1 makes it impossible. I wonder if this is a mistake in my understanding or if the algorithm is written wrong.
4. I cannot get the information that tried to be conveyed by Fig. 3. The effect of the bidirectional filtering is not clear in the figure. More detailed captions are required.
5. Though the proposed method is demonstrated to be effective, the whole pipeline is too complex. Though the pipeline is training-free, 3 pre-trained models are required. Thus, their mistakes and shortcomings are also inherited by the proposed method. Even worse, before the segmentation begins, text prompts are required for computing the segmentation logits. This implies the user must know the categories of the objects in the scene. In contrast, similar methods that only use SAM for segmentation do not have such limitations.
6. The hybrid pipeline also leads the concerns about the segmentation efficiency. As shown in Table 11, the inference speed is very slow. A time consumption comparison with the other existing methods is required.

**Questions:**

Please see the weaknesses that the writing is not clear. I have many difficulties in understanding the proposed method.

---

### Note · Authors · 2024-11-15

I have read and agree with the venue's withdrawal policy on behalf of myself and my co-authors.